# Efficacy of heparin in respiratory support of near-term rabbits with meconium-induced acute lung injury: Linear regression model analyses

Siyu Xie[1], Qiang Gu[1]\*, Guiyin Zhuang[1], Xiaojing Guo [2], Bo Sun[2]\*

**1** Department of Pediatrics, The First Affiliated Hospital of Shihezi University, Shihezi, Xinjiang Uyghur Autonomous Region, China, **2** Department of Pediatrics, Children's Hospital of Fudan University, National Children's Medical Center, The Laboratory of Neonatal Diseases of National Commission of Health, Shanghai, China

\* guqiang106@sina.com (QG); bsun@shmu.edu.cn (BS)

## Abstract

### Objectives

To explore the pharmacotherapeutic efficacy of heparin in the management of meconium-induced acute lung injury (ALI) in near-term newborn rabbits subjected to mechanical ventilation (MV) and ancillary respiratory medications.

### Methods

Newborn rabbits at 30-day gestation (term 31 days) were anesthetized, intratracheally intubated and received human meconium-saline suspension, followed by parallel MV with individually adjusted tidal volume in a multi-plethysmograph-ventilator system. When ALI was induced after initial 3-h MV, therapeutic effects of single or combined subcutaneous heparin (100 U/kg), surfactant (200 mg/kg), and inhaled nitric oxide (iNO, 10 ppm), were compared for lung protective ventilation and survival as outcome, analyzed with linear regression models.

### Results

Significantly reduced respiratory compliance by meconium was reinstalled during ensuing 7-h MV, with improved survival, among the treatment groups. The impact was verified by lung injury severity, surfactant phospholipid pools, and multiple mRNA expressions of surfactant proteins, lung fluid clearance-related factors, inflammatory mediators, growth factors, endothelial cell injury and coagulation-related factors as subphenotyping biomarkers. The overall benefits of heparin alone, or exerted with the dual and triple regimens, were discernible by both generalized linear model and Cox proportional hazard ratio regression for survival and other major variables as outcome. Its adverse effects were intangible.

**Data availability statement:** All relevant data are within the manuscript and its Supporting information files.

**Funding:** This research was supported by National Natural Science Foundation of China (NSFC, No.82360315, Q.G.). The foundation (NSFC) had no influence in the study design, performance, analysis, and interpretation of experimental data and in writing the manuscript.

**Competing interests:** The authors have declared that no competing interests exist.

## Conclusion

The comparable efficacy of heparin, alongside the PS and NO, was corroborated in attenuating meconium-mediated, ventilator-induced ALI, which should warrant clinical investigation to validate.

---

## Introduction

Meconium aspiration syndrome (MAS) is a rare but fatal perinatal comorbidity in full-term or post-term infants. Pathogenesis and pathophysiology of MAS are related to airway obstruction and alveolar impairment-associated derangement of lung mechanics, persistent hypoxemic pulmonary vascular restriction, and associated ventilation-perfusion mismatching, complicated with persistent pulmonary hypertension of the newborn (PPHN) [1–4]. The standard care for MAS-associated PPHN includes inhaled nitric oxide (iNO) as a selective pulmonary vasodilator, even though some very critically ill PPHN infants may require extracorporeal membrane oxygenation (ECMO) as life support to survive [5]. Moreover, access to advanced supportive therapies, including ECMO, is often hampered by availability and affordability in resource-limited services for neonatal critical care. Therefore, exploring more cost-effective and readily available ancillary medications for mechanical ventilation (MV) in critical care should be imperative.

Pathophysiologically, MAS induces inflammation and oxidative cascade reactions in the lungs through hyperoxia, MV and chemical pneumonia-associated damage [6]. Meconium per se may inhibit endogenous pulmonary surfactant (PS) activity and metabolism, thus causing impairment in lung mechanics and other physiological functions [2,7,8], and endogenous surfactant phospholipid synthesis [9]. Various surfactant preparations are routinely used for the treatment of respiratory distress syndrome (RDS) in preterm infants with concrete benefits and safety for survival [10]. In term and post-term newborn infants with MAS, past studies suggest that exogenous PS may augment conventional supportive therapies, and its off-label use in routine care is debatable in the reduction of death in severe MAS [1–3]. We consider the initial lung impairment of MAS should fall into the classic definition of acute lung injury (ALI), or nowadays as an early stage, mild in severity, of acute respiratory distress syndrome (ARDS) in neonatal and pediatric patients [11]. Surfactant is often off-label used in term neonates with MAS, and infants and young children with infection (pneumonia, sepsis)-associated ARDS-like respiratory failure [12–15]. Solid evidence confirms that iNO improves oxygenation and reduces the need for ECMO in term and near-term neonates with PPHN [16,17]. And, even in the early pioneering iNO trials, e.g., iNO reduced the severity of PPHN in infants with persistent hypoxemic respiratory failure was with prior PS treatment failure, with reduced ECMO dependence, but not altered the sole risk of neonatal death [5]. Because iNO is expensive and requires a specific facility to implement, it is our particular interest to explore alternative drugs in neonatal critical care in resource-limited conditions.

Unfractionated heparin (UFH) is a commonly used anticoagulant that binds to antithrombin-III (AT-III) to form a heparin-AT-III complex, thereby enhancing the effects

of AT-III through inactivating coagulation factors IIa, XIIa, XIa, IXa, and Xa, and inhibiting the conversion of fibrinogen to fibrin [18]. The latter is also known as a surfactant inhibitor [19,20]. Heparin also inhibits platelet adhesion and aggregation, thereby preventing platelet integration and release of procoagulant factors [21]. UFH is used in the treatment of hemodynamic disorders, exerts potential in growth factor modulation, and anti-inflammation, anti-viral infection, and anti-metastasis, in addition to anti-thrombotic activity [21,22]. The role of heparin in angiogenesis and organ growth remains inconclusive, with conflicting data reported in the literature. A study indicated that high-dose intraperitoneal heparin (250 and 500 U/kg) interferes with vascular endothelial growth factors (VEGF) signaling pathway, inhibiting compensatory lung growth following left pneumonectomy in mice [23]. While another study revealed that intraperitoneal administration of heparin (250 U/kg) suppresses the formation of neutrophil extracellular traps by attenuating histone toxicity, thereby improving alveolarization and pulmonary vascular development in newborn rats with hyperoxia-induced BPD. However, high-dose heparin (500 U/kg) shows no significant effect [24]. A multicenter, randomized controlled, double-blind, phase 3 trial demonstrated that nebulized heparin treatment for patients at high risk of, or with confirmed ARDS (age > 18 years), reduced the progression of the lung injury and shortened the hospital stay [25]. The UFH, as well as low molecular weight heparin (LMWH), are used in daily NICU practice with adequate indication and criteria [26]. Its efficacy in neonatal hypoxemic respiratory failure has not been investigated yet. We assumed that there should be pathogenesis of pulmonary vascular restriction together with hypercoagulability, secondary to hypoxia and hypoxemia in MAS. We consider it to be rational in the current experiment testing a single dose of 100 U/kg UFH in a well-established near-term rabbit MAS model [26,27], which facilitates a concurrent use of PS and iNO. We hypothesized that, following the lung-protective mode of MV protocol, the combination of the two or three medications should be superior to their single use in the improvement of lung mechanics, and their mechanisms in respiratory physiological, pathophysiological and pharmacotherapeutic action may be explained.

## Materials and methods

### Animals, meconium, and surfactant

The study protocol was approved by the Ethics Committee of Children's Hospital of Fudan University (No.2023349). Twenty-one healthy pregnant New Zealand White rabbits of 30 days (term 31 days) of gestation were provided by Shanghai Songlian Experimental Animal Center [27,28]. The day before the experiment, these rabbits were brought to the experimental site and provided with sufficient food, water, and appropriate shelter. A total of 205 newborn rabbits were delivered by cesarean section in does with a series of anesthesia, face mask oxygen supply, and temperature maintenance in the experiment according to Chinese regulations for experimental animal care in medical research [27,28].

Fresh, dark-greenish, meconium samples from the same batch were collected from full-term healthy newborns on their first postnatal day, well mixed with sterile water, lyophilized and grounded as powder, and stored at -20°C until use [27]. On the day of experiment, a suspension of 100 mg dry weight/ml sterile saline was prepared, and a volume of 4 ml/kg body weight was administered intratracheally to the experimental animals immediately at birth, followed by MV [7,8,27]. The pulmonary surfactant (PS) was prepared through fresh porcine lung lavage with normal saline, then subjected to a series of high-speed, gradient centrifugation, chloroform/methanol extraction, and cold acetone precipitation. The dried material was mixed with sterile saline at a concentration of 80 mg phospholipids/ml as a suspension [29].

### Animal management and MAS model

On the day of experiment, 2 ml of diazepam (Shanghai Xudong Haipu Pharmaceutical Co., Ltd., Shanghai, China) and 10 ml of 20% urethane (Ethyl carbamate, BBI Life Sciences, Shanghai, China) were injected intramuscularly sequentially into the pregnant rabbits. When they were sedated and anesthetized, additional urethane (15–20 ml) was administered intravenously for maintenance. A fraction of pure oxygen ($FiO_2$ 1.0) was administered via a face mask throughout the initial one-hour delivery of offspring. Local anesthesia with subcutaneous infiltration of lidocaine (Shandong Hualu Pharmaceutical Co. Ltd., Jinan, Shandong, China) was followed by cesarean section and fetal rabbits were removed from the uterus

in sequence, dried, and their birth weight (BW) was weighed [27,28]. After the completion of delivery, the pregnant rabbits were euthanized by intravenous injection of an overdose of potassium chloride.

The newborn rabbits were injected intraperitoneally with 0.15 ml of lidocaine and 10% glucose mixture (vol/vol 1:1), followed by subcutaneous infiltration of the anterior neck with the same lidocaine, when needed, tracheotomy and intra-tracheal intubation with a stainless steel canula. Then, 4 ml/kg of meconium suspension was injected through the tracheal cannula, while sham control pups were injected with an equal volume of sterile saline. Then pups were connected to a multi-plethysmograph-ventilator system that allows up to 12 animals to be ventilated simultaneously [27–31]. This device consists of a Servo ventilator (900C, Siemens-Elema, Solna, Sweden), a pneumotachometer (RSS100-HR, Hans Rudolph Inc., Kansas City, KA), a pressure transducer (Shanghai Yangfan Electronic, Shanghai, China), and a bioelectric signal amplifier. The ventilator was set to pressure-controlled mode, and the target tidal volume was maintained at 4–6 ml/kg by adjusting the peak inspiratory pressure (PIP) with a ventilation rate of 40 breaths/min, an inspiratory/expiratory ratio of 1:2, $FiO_2$ 1.0, and positive end-expiratory pressure (PEEP) of 2–3 cmH$_2$O [27–32]. Respiratory physiologic variables were monitored and recorded by an automated physiologic monitoring system (PowerLab, ADInstruments Pty Ltd, New South Wales, Australia). Pups were connected to the ventilator, and MV was initiated at time 0. Intraperitoneal injections of 0.1–0.15 ml of a mixture of 10% glucose, 5% NaHCO$_3$, and 2% lidocaine (vol/vol/vol 5:3:2) were given to each pup every 1.5–2 h. During the 10-hour ventilation period, the animal body temperature was maintained with a heating pad [27]. They were all sacrificed by intracranial injection of 0.5 mL 2% lidocaine to cause immediate cardiac arrest and termination of MV [27,28].

## Experimental phases and drug regimens

**Phase I: Non-iNO treatment regimen.** In this phase, after initial meconium (M) instillation and approximately 3 h of MV to attain severe hypoxemia and respiratory failure, the animals were divided into five treatment groups according to protocol, randomly receiving (1) M, saline (2.5 ml/kg); (2) MS, intratracheal injection of 200 mg/kg, 2.5 ml/kg surfactant phospholipid-saline suspension; (3) MH, subcutaneous injection of UFH (100 U/kg, SPH First Biochemical & Pharmaceutical Co. Ltd., Shanghai, China); (4) MSH, a combination of surfactant and UFH with the same dosage as in (2) and (3); (5) C, no meconium but intratracheally 2.5 ml/kg normal saline. They were subjected to continued MV until due time or terminated MV due to early death with irreversible bradycardia. An additional non-MV group was assigned within the littermates sacrificed immediately after birth (time 0, C0), and their lungs were used as references for lung morphometry and biological assays (qPCR, see below).

**Phase II: iNO treatment regimen.** In this phase, there were four groups receiving meconium (M) as above in phase I. iNO was given through the ventilator circuit at a dose of 10 parts per million in volume (ppm) to all the pups throughout MV. The rest of ventilator settings were the same. Thus, the four groups were designated as MN, MSN, MHN, and MSHN, where N stands for iNO. There was no control with MV only (C), but the MN instead. Inhaled nitric oxide (iNO) was provided by a high-purity (> 99.999%) nitrogen gas dilution as a mixture, at a stock concentration of NO of 1000 ppm, and injected continuously into the inspiratory limb of ventilator circuit, through an automated gas mass controller, and monitored by electrochemical sensors for NO and NO$_2$ [29].

To sum up, there consisted of seven groups in the two phases for intergroup comparisons of the pharmacological actions, i.e., single (MS, MH, MN), dual (MSH, MSN, MHN), or triple combination (MSHN) of surfactant (S), UFH (H) and iNO (N) (Fig 1). All the animals were assigned by inter-litter (N vs. non-N), or intra-litter (H or S) administration in random order with a minimum of 20 animals per group to ensure subsequent lung samples (≥6) for sub-group histopathological or biochemical measurements.

## Measurement of lung mechanics and survival status

Lung mechanic parameters (PIP, PEEP, Vt) were frequently measured and recorded at 15, 30, 45, 60, 90 and 120 min of MV, and then recorded every hour until 10 h or early death due to bradycardia (< 30 beats/min on ECG), along with

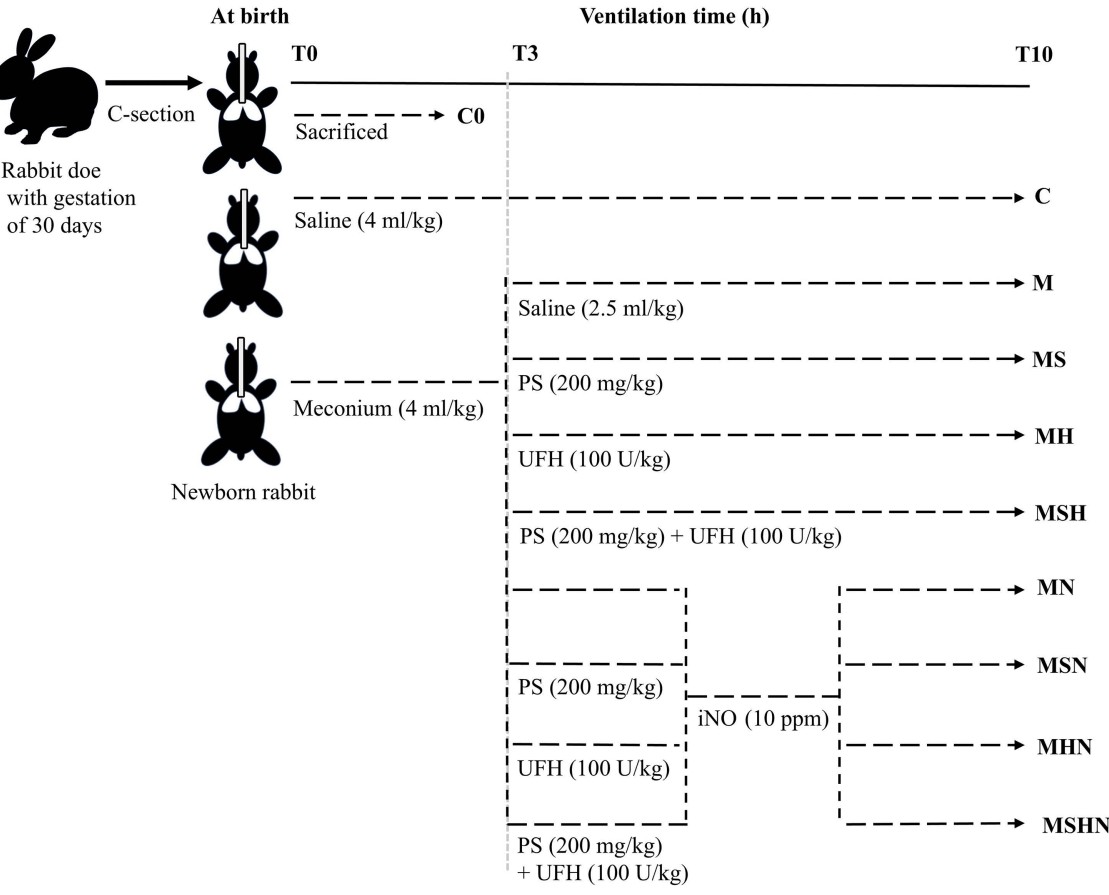

**Fig 1. Flow chart of the experiments.** Group definitions: C0, neither meconium (M) nor mechanical ventilation (MV); C, no M but with MV; The following eight groups had M at birth and were subjected to parallel MV for 3 h (to induce lung injury), then treated with rescue drugs and MV for additional 7 h: M, saline only; MS, surfactant (S) only; MH, heparin (H) only; MSH, both S and H; MN, inhaled nitric oxide (N) only; MSN, both S and N; MHN, both H and N; MSHN, combined S, H and N.

cyanotic and pale appearance [29]. Dynamic compliance of respiratory system (Cdyn, ml/cmH$_2$O/kg) at each time point was calculated by Vt/(PIP-PEEP)/BW. A mean value of Cdyn (Cdyn$_{mean}$) over the late 7 hours of ventilation after 3-h MV and ALI was used for the generalized linear models and Cox proportional hazard ratio regression model (see below statistics). The survival time of all animals was recorded. Pneumothorax (PTX) and sex were visually identified at autopsy.

## Lung sample processing

Left ventricular blood was taken for mixed blood gas analysis. Both lungs of each group of animals were randomly assigned for histopathologic or biochemical analysis. The accessory lobe of right lung was resected, and the wet-to-dry weight ratio (W/D) was determined by weighing before and after placing it in a heated oven at 60 °C for at least 48 h to estimate the fluid content of the entire lung tissue.

## Lung histopathology

The lungs of pups (7–8 per group) used for histopathologic measurements were fixed by bilateral lung perfusion. Simultaneously, a constant pressure of 30 cmH$_2$O was provided by the ventilator through the intratracheal tube for 1 minute,

followed by a decrease to 10 cmH$_2$O as a deflation pressure to maintain continuous alveolar expansion (assumed functional residual volume). Lung tissue was perfusion-fixed through 4% paraformaldehyde, then immersed in 4% formalin for three days. The lung sample was embedded in paraffin, sectioned, and stained with hematoxylin-eosin for further semi-quantitative assessment of alveolar expansion (Vv), coefficient of variation of Vv [CV (Vv)], and lung injury score (LIS), as previously described [28,30,31]. Lung injury scores are graded from 0 to 4 based on the severity of edema, hemorrhage, inflammatory cell infiltration, small airway injury, and meconium distribution: 0 indicates no lesions; 1 indicates mild and localized lesions (<25%); 2 indicates moderate and localized lesions (25%–50%); 3 indicates moderate but extensive or locally prominent lesions (50%−75%); 4 indicates severe lesions (>75%). Additionally, the periodic acid-Schiff stain was performed to observe the distribution of meconium in the lungs.

## Ultrastructural morphology of the lungs

Freshly excised lung tissue samples of 1 mm$^3$ size from the middle lobe of the right lung were fixed in glutaraldehyde for 24 h at 4°C, followed by dehydration, embedding, sectioning and staining according to standard procedures, and examined by transmission electron microscopy (FEI Tecnai G2 Spirit TWIN, ThermoFisher Scientific, Boston, MA, USA).

## Biochemical analysis

For the animals scheduled for biochemical analyses (10–13 per group), the left lung was lavaged three times with saline at 15 ml/kg, BAL fluid (BALF) was pooled, and the volume was measured and recorded. The post-lavage lung tissue was homogenized with saline. The collected BALF fluid was immediately centrifuged at 4°C at 2000 rpm for 15 min to remove cell debris, and the supernatant was frozen at -20°C [30,31]. Total phospholipids (TPL) and disaturated phosphatidylcholine (DSPC) were measured in BALF and lung tissue homogenate (LH) according to the methods with correction by BALF volume and BW [31,33,34]. Total proteins (TP) in BALF were quantified by BCA kit (Pierce BCA Assay Kit 23225, Thermo Scientific, Rockford, IL). These variable values were also presented as two-sided lungs by multiplying by 1/0.45.

## Quantification of mRNA expression by quantitative PCR (qPCR)

The rest of the right lung tissue (6−13 per group) was used to measure mRNA expression by qPCR analysis by Light-Cycler 480 II (Roche, Basel, Switzerland) [27,29]. Sequence information of target mRNA genes was obtained from the nucleotide database (www.ncbi.nlm.nih.gov/gene/). The target mRNA included: (1) surfactant proteins (SP-A, B, C, D); (2) growth factors (GFs): insulin-like GF (IGF-1, IGF-2), vascular endothelial GF (VEGF), keratinocyte GF (KGF); (3) inflammatory mediators: toll-like receptor (TLR)-2, TLR-4, myeloid differentiation factor 88 (MYD88), nuclear transcript factor kappa B (NF-κB, subunit p50), tumor necrosis factor (TNF)-α, interleukin (IL)-1β, IL-6, IL-8; (4) endothelial nitric oxide synthase (eNOS) and inducible nitric oxide synthase (iNOS); (5) type I alveolar epithelial cell marker and lung fluid clearance-related factors: aquaporin (AQP)-5, Na$^+$-K$^+$ ATPase α1; (6) endothelial cell proliferation, injury, and coagulation-related factors: angiopoietin (Ang-1, Ang-2), Tie-1, Tie-2, syndecan-1, tissue factor (TF). The primers were designed by Sangon Biotech (Shanghai, China), and the primer sequences are summarized in the supplemental S1 Table. The amplifying reaction was carried out in a 10 μL volume containing 5 μL SYBR Green Pro TaqHS Premix (Accurate Biotechnology Hunan Co. Ltd, Changsha, China). The mRNA expression of each gene was normalized to β-actin, and the average fold change was calculated using the ΔΔCT method, in reference to that of the C0 group.

## Statistical analysis

All statistical comparisons were performed by SPSS 23.0 (IBM, Armonk, NY), and figures were produced by GraphPad Prism 9.5.1 (La Jolla, CA). Using Groups C and M as controls, the three groups without iNO and the four groups with iNO were statistically analyzed against the control groups. Specifically, statistical analysis was separately performed for the C, M, MS, MH, and MSH, and the C, M, MN, MSN, MHN, and MSHN groups. Kaplan–Meier analysis and multivariate

Cox regression model were used for survival analysis. Comparisons between groups were made using the log-rank test. Continuous variables are expressed as mean ± standard deviation (SD), or median and interquartile range (IQR), or range (minimal to maximum for LIS). One-way ANOVA (F) or Kruskal-Wallis (H) tests were performed for differences among the groups, followed by Bonferroni post hoc test or Mann-Whitney U test for between-group comparison, respectively. Proportional data are presented as numbers (n) and percentages (%), and analyzed by Pearson Chi-square test. A $p$-value < 0.05 was considered statistically significant for the difference. Additional analysis for drug synergistic action was performed by combining all the treatment groups containing one of the three drugs, e.g., UFH but with the other two drugs (i.e., PS and iNO) evenly allocated as all-UFH in the four heparin-containing groups (MH, MSH, MHN and MSHN), in comparison with the non-UFH (M, MS, MN, MSN, and so forth for the all- or non- PS and iNO combinations, respectively, which distinguishes them from the original treatment groups (using H, S, N). The eight groups established via the meconium instillation model were coded according to the drugs administered, using a binary scheme (1 for drug presence, 0 for drug absence), e.g., code 1,1,1 referring to the MSHN group. These data were analyzed using generalized linear models (GLMs). The independent variables included PS, UFH, and NO, along with their two-way and three-way interaction terms. The dependent variables were survival outcome, $LIS_{total}$, Vv, CV (Vv), $DSPC_{BALF}$, pH, $PCO_2$, LAC, W/D, and $Cdyn_{mean}$. Post-hoc pairwise comparisons were adjusted using the Bonferroni method.

## Results

### General conditions

A total of 205 GA 30-day full-term newborn rabbits were included in this experiment, with a median (IQR) of 49 (40, 56, range 23–78) g. There was no significant difference in the rate of sex or pneumothorax among the groups (Table 1). After 10 h of ventilation, in comparison with the C group, the M group had significantly lower survival rate. Compared with the M group, the 10 h survival rates of those receiving three drug-treated (H, S and N) were all increased, with MSHN being the same as that of C (100%). The survival rates of pooled data analysis for each of the three drugs as the prevalence treatment group in the PS and iNO treated groups showed a tendency to increase compared to respective non-PS or non-iNO group. Compared to the non-UFH group, the 10 h survival rate of those in the UFH-treated group increased by about 18% (Table 1). In those exposed to meconium, W/D values of the MSHN and MN had 16% respective 10% reduction, compared to the M (Table 1).

### Blood gas analysis

Blood gas values were improved in the three drug-treated groups, with the iNO-treated groups showing the most pronounced improvement, though hypercapnia, acidosis, and hyperlactacidemia remained. In the pooled treatment groups, all-PS and all-iNO had significantly higher pH and lower $PCO_2$ compared with respective non-PS and non-iNO groups (S2 Table).

### Survival status

There was no significant difference in overall survival rates among the treatment groups (Fig 2). By log-rank test, it indicates that the survival rates of MH, MN, MSH, MHN and MSHN groups were significantly better than that of the M group, but not for the MS and MSN, with C and MSHN having the most significant overall survival benefits. The pooled treatment group data showed that all-UFH was associated with prolonged survival time (S1 Fig).

### Measurement of lung mechanics

In group C, the initial Cdyn was around 0.5 ml/cmH$_2$O/kg, rising to around 0.7 ml/cmH$_2$O/kg during MV, whereas in group M, the Cdyn gradually declined, reaching the lowest level around 0.4–0.5 ml/cmH$_2$O/kg. All the animals receiving meconium had decreased Cdyn than the C group and persisted with >30% reduction after 3 h of ventilation, which was

                                                                                                    

**Table 1. Basic data and survival rate among different treatment groups with or without meconium induced ALI.**

| Group | n | BW | Male | Survival-10 h | PTX | W/D | LIS$_{total}$ |
|---|---|---|---|---|---|---|---|
| | | g | n (%) | n (%) | n (%) | mean ± SD (n) | median (min-mix) |
| A. Treatment groups | | | | | | | |
| C0 | 21 | 48.7 ± 11.6 | 9 (43) | | 0 (0) | 6.30 ± 1.09 (20) | 1.0 (0-2) |
| C | 21 | 53.3 ± 8.5 | 8 (38) | 21 (100)$^{§§}$ | 0 (0) | 6.23 ± 0.82 (18) | 2.0 (0-3) $^{§§§}$ |
| M | 21 | 47.6 ± 13.2 | 10 (48) | 14 (67) | 2 (10) | 6.21 ± 1.16 (19) | 7.0 (5-11) |
| MS | 20 | 50.3 ± 9.2 | 10 (50) | 17 (85) | 1 (5) | 5.88 ± 1.07 (19) | 5.0 (3-9) |
| MH | 20 | 54.3 ± 10.6 | 9 (45) | 19 (95)$^{§}$ | 1 (5) | 5.85 ± 0.99 (19) | 6.5 (3-7)$^{^^}$ |
| MSH | 20 | 53.2 ± 11.6 | 10 (50) | 19 (95)$^{§}$ | 0 (0) | 5.80 ± 0.85 (19) | 5.0 (3-6) |
| MN | 20 | 46.8 ± 10.6 | 9 (45) | 19 (95)$^{§}$ | 0 (0) | 5.56 ± 0.83 (18) | 4.0 (3-5) |
| MSN | 20 | 45.1 ± 10.5 | 12 (60) | 17 (85) | 0 (0) | 6.03 ± 0.93 (20) | 4.0 (2-5) $^{§}$ |
| MHN | 21 | 42.9 ± 11.4 | 9 (43) | 20 (95)$^{§}$ | 1 (5) | 5.63 ± 1.12 (21) | 3.5 (3-7)$^{§}$ |
| MSHN | 21 | 46.1 ± 9.0 | 9 (43) | 21 (100)$^{§§}$ | 0 (0) | 5.19 ± 0.75 (16)$^{^§}$ | 4.0 (3-5)$^{^}$ |
| [Total | 205 | 48.8 ± 11.1 | 95 (46) | 167 (81) | 5 (2) | 5.88 ± 1.01 (189)] | |
| B. Pooled treatment groups | | | | | | | |
| all-PS | 81 | 48.6 ± 10.4 | 41 (51) | 74 (91) | 1 (1) | 5.75 ± 0.95 (74) | 4.5 (2-9) |
| non-PS | 82 | 47.8 ± 12.0 | 37 (45) | 72 (88) | 4 (5) | 5.81 ± 1.05 (77) | 5.0 (3-11) |
| all-UFH | 82 | 49.0 ± 11.5 | 37 (45) | 79 (96)$^{**}$ | 2 (2) | 5.63 ± 0.96 (75) | 5.0 (3-7) |
| non-UFH | 81 | 47.4 ± 10.9 | 41 (51) | 67 (83) | 3 (4) | 5.93 ± 1.02 (76) | 5.0 (2-11) |
| all-iNO | 82 | 45.2 ± 10.3$^{***}$ | 39 (48) | 77 (94) | 1 (1) | 5.63 ± 0.96 (75) | 4.0 (2-7) $^{***}$ |
| non-iNO | 81 | 51.3 ± 11.4 | 39 (48) | 69 (85) | 4 (5) | 5.93 ± 1.02 (76) | 6.0 (3-11) |

Values are means ± SD. Group definitions: C0, no meconium and no mechanical ventilation (MV); C, no meconium but with MV; Meconium (M) exposed groups with drug treatment and parallel MV: M, meconium only, saline-treated; MS, surfactant (S) only; MH, heparin (H) only; MSH, both S and H; MN, inhaled nitric oxide (N) only; MSN, both S and N; MHN, both H and N; MSHN, combined S, H and N. Definitions of pooled rescue drug treatment groups: e.g., all-PS or non-PS, all with or without S, i.e., MS + MSH + MSN + MSHN vs. M + MH + MN + MHN, respectively, and so forth for all- or non-UFH (H) and all- or non-iNO (N); BW, birth weight (g); n, number of newborn pups; PTX, pneumothorax; W/D, wet-to-dry lung weight ratio. $^{^}p < 0.05$, $^{^^}p < 0.01$ vs. C; $^{§}p < 0.05$, $^{§§}p < 0.01$, $^{§§§}p < 0.001$ vs. M. $^{**}p < 0.01$, $^{***}p < 0.001$ vs. corresponding pooled non-designated drug group.

modestly improved (by 10–20%) in all three drug-treated groups. The values of Cdyn of the treated groups at each time point were not significantly different (Fig 2). Changes of Cdyn and survival rate over the whole MV period, and blood gas values at the end of MV, of all the groups are shown in the supplemental materials (S2 and S3 Tables)

## Lung histopathology and morphometry

Representative photomicrographs of the lung sections from each group are shown in Fig 3. By morphometric measures, Vv increased and CV (Vv) decreased in all the drug-treated groups compared with the M. The MSN and MSHN had the most significant improvement, which was superior to that of MN (S4 Table). The LIS$_{total}$ values were higher in the M compared to the C group, along with higher CV(Vv). The LIS$_{total}$ decreased in all the drug-treated groups compared to the M, with the most pronounced decrease in the four iNO-treated groups (Table 1). The iNO-treated groups had milder inflammation and bronchiolar epithelium disruption compared to the other two drug-treated groups (S5 Table). The distribution score of meconium indicated that meconium was uniformly distributed within the alveoli and small airways across all meconium groups (S5 Table). Pooled treatment group data showed that the all-iNO group had significantly low score values of inflammation, bronchiolar epithelium disruption, and LIS$_{total}$, compared with non-iNO (Tables 1, S5); Vv was significantly higher and CV (Vv) was significantly lower (S4 Table). Compared with non-PS, Vv significantly increased and

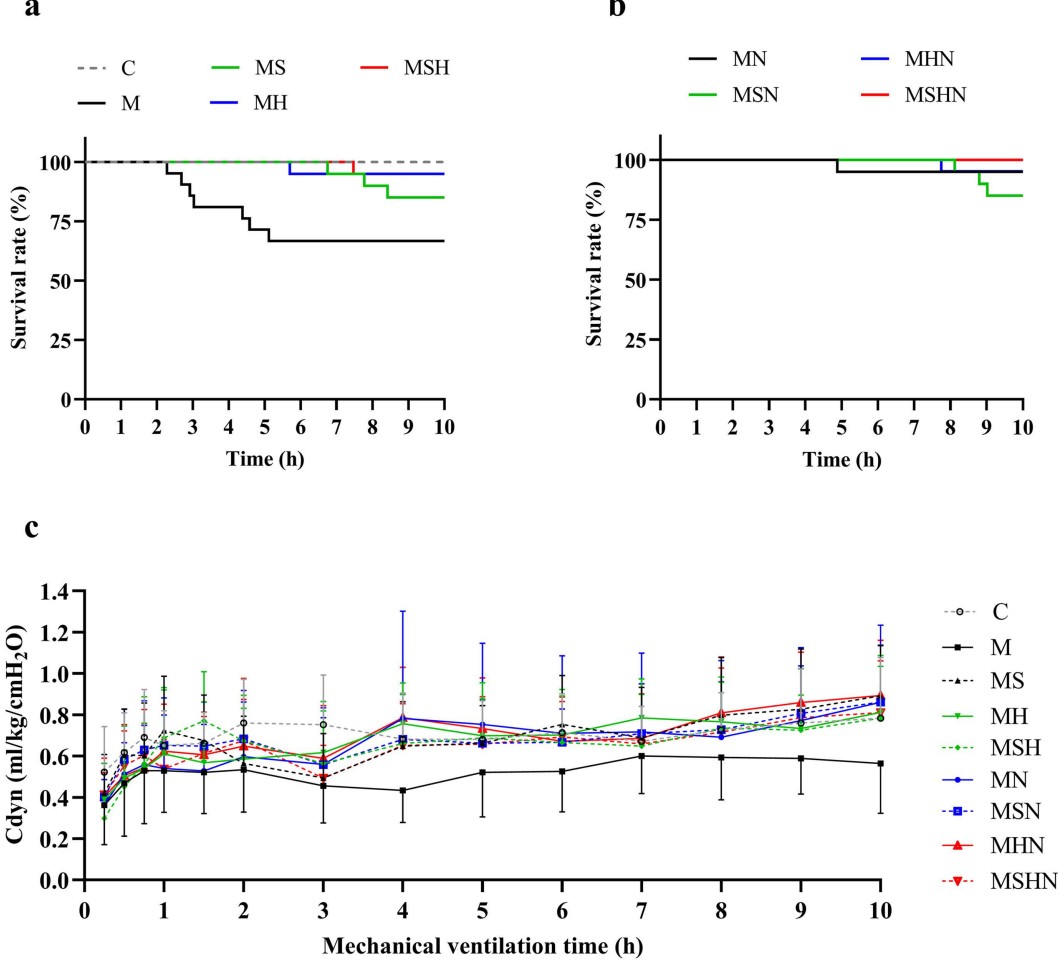

**Fig 2. Trend of survival and lung mechanics. a**, **b** Kaplan-Meier survival curves. **c** Trend of the dynamic compliance of respiratory system (Cdyn) during ventilation. For group definitions, see Table 1 legends. Data are presented as means±SD in **c**, n=20-21 in **a**, **b** and **c**.

CV (Vv) decreased in the all-PS group. There was a trend towards increased Vv and decreased CV(Vv) in the all-UFH compared to the non-UFH group, but there was no statistical significance (S4 Table).

## Lung ultrastructure observation

By transmission electron microscopic observation, there were abundant capillaries covered by type I alveolar epithelial cells in the alveolar septa, along with abundant lamellar bodies and tubular myelin formation, readily identifiable in alveolar space, consistent with the characteristics of full-term neonatal rabbits at their early alveolar stage of development. Formless material, supposed as meconium, was seen in the alveolar space and small airway lumen of all meconium-exposed lungs. Well-parallelly lined phospholipid layers, or cross-linked tubular myelin, were readily identifiable at high magnification in the C group. This feature was hardly discernible in the M group. There were large multilayer, ring-shaped, membrane-like materials adjacent to type II alveolar epithelial and interstitial cells in the PS-treated, MSHN group, a special feature not found in other PS-treated or non-PS groups (Fig 4).

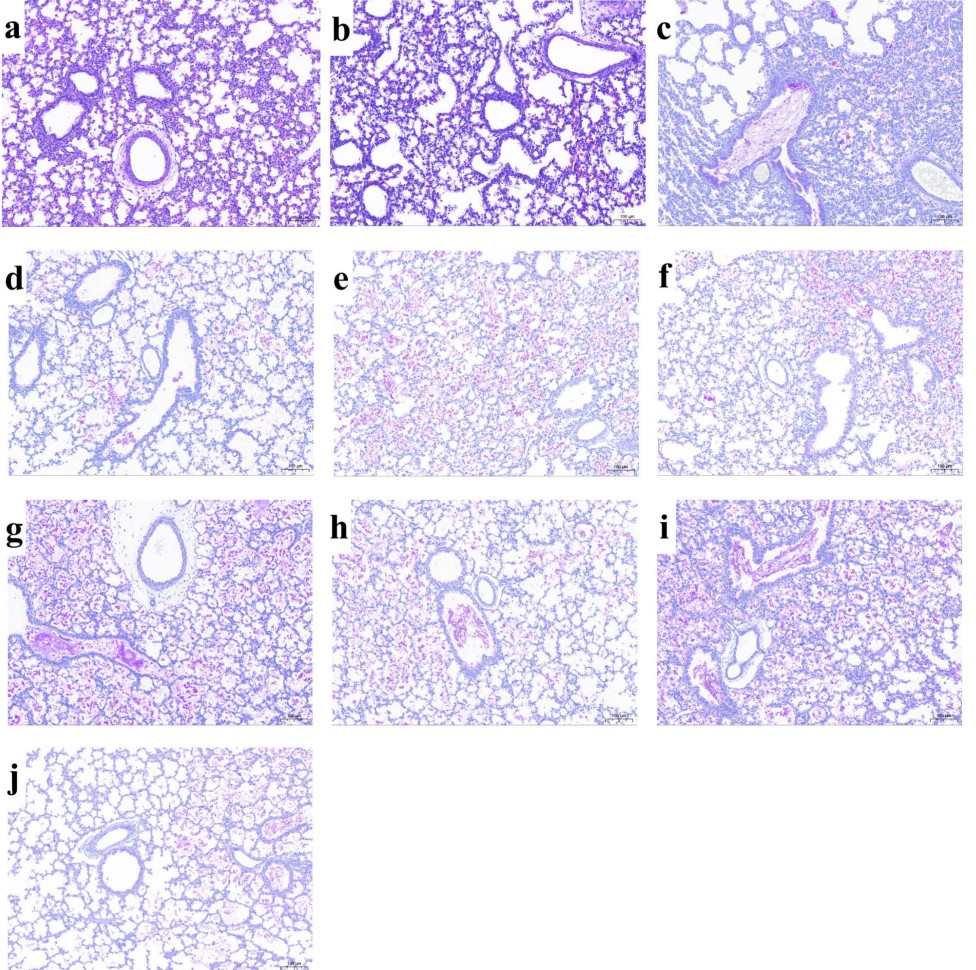

**Fig 3. Photomicrographs of newborn rabbit lung sections at the end of ventilation. a-j** in order: C0, C, M, MS, MH, MSH, MN, MSN, MHN, MSHN. Group definition: see Table 1 legends. **a, b**: Hematoxylin-eosin stain; **c-j**: Periodic acid-Schiff stain; bar scale = 100 µm.

## Lung biochemical measurements

Both TPL and DSPC in BALF were lower in the M than in the C group, by 61% and 59%, respectively. Compared with M, the MS had 161% in TPL and 151% DSPC increment (Table 2). In LH and LH + BALF as the total pool, the PS-treated groups had higher TPL, DSPC, and DSPC/TPL (S6 Table). By pooled treatment group data, the all-PS was associated with increases in TPL, DSPC and DSPC/TPL in BALF, LH, and total pool. TP was increased and DSPC/TP decreased in the M compared to the C group. After the three-drug treatment, DSPC/TP was increased compared to M, especially in the PS-treated groups (Table 2). The values of TPL in the total metabolic pool did not change significantly between groups, so the relative rates of change of DSPC and DSPC/TPL in each group were compared to reflect the effect of treatment on phospholipid content. By comparing the rate ratio of the values of DSPC/TPL between each group, it was 1.13 in MS vs. M, and 1.65 in MSH vs. MH, showing that UFH augmented the surfactant pool. Similarly, the rate ratio of DSPC/TPL of MSN to MN was 1.40, also indicating a synergistic effect of iNO and surfactant. The rate ratio of DSPC/TPL of MN to M was 0.84, and that of MHN to MH was 1.18, again suggesting that UFH augmented the effects of iNO on the surfactant phospholipid pool (S7 Table).

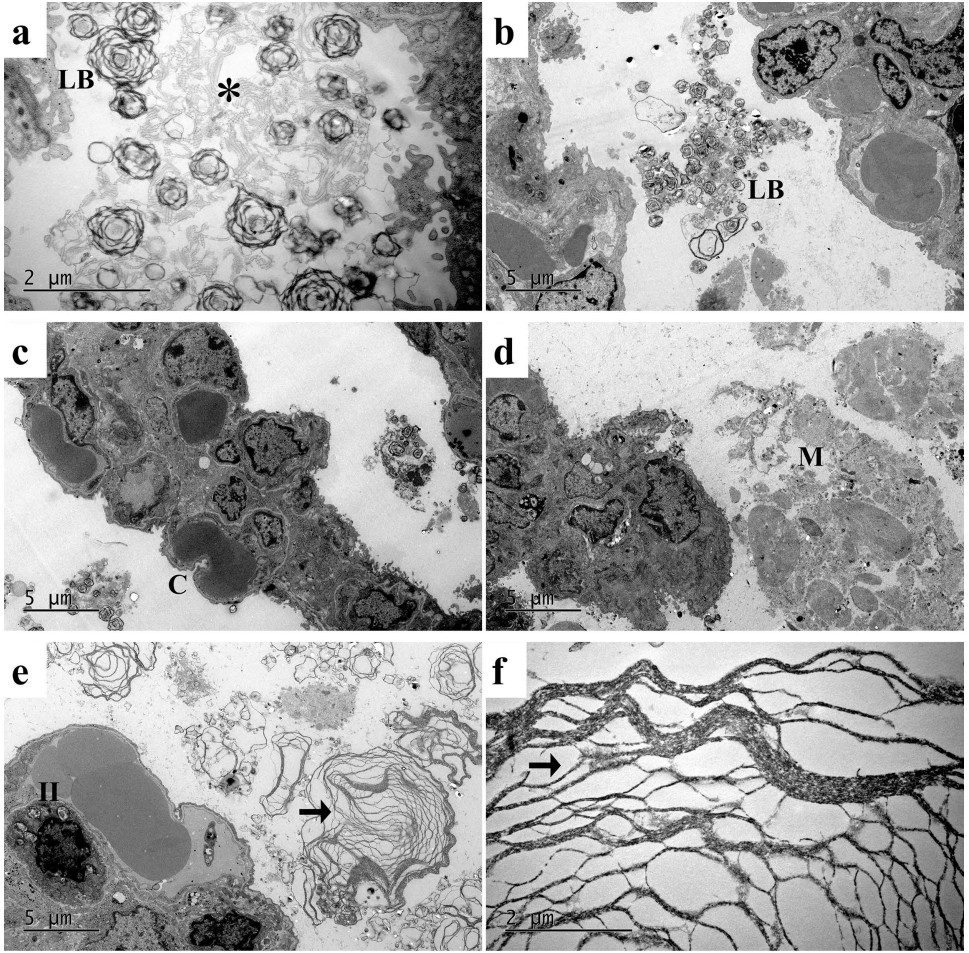

**Fig 4. Ultrastructure of alveoli through transmission electron microscope. a**, group C; **b**, MH; **c**, MN; **d**, MSN; **e** and **f**, MSHN. For group definition see Table 1 legends. **b**, **c**, **d**, **e**: scale bar = 5 μm; **a**, **f**: scale bar = 2 μm. LB, lamellar body; * tubular myelin; C, capillary; M, meconium; II, type II alveolar epithelial cell; →surfactant phospholipid layers.

## Measurement of mRNA expression

The magnitude of mRNA expression among the groups varied without significant difference, with that of IL-1β being 2−4 fold on average in all eight meconium-exposed groups and C than the C0 (S2–S4 Figs). The Ang-1 and Ang-2 levels in the treatment group showed a trend toward elevation compared with the M group, but no significant difference was observed. And expression of Tie-1 was decreased in the treatment group compared with Group M, but no significant difference was observed; whereas that for Tie-2 was not significantly altered. Nonetheless, all-iNO had significantly lower mRNA expression of TLR-2, MyD88, NF-κB, TNF-α, IL-1β, IL-6, IL-8, Na$^+$-K$^+$ ATPase α1, IGF-2, VEGF, and eNOS than the non-iNO groups, whereas all-UFH had significantly lower TF mRNA expression than the non-UFH in the pooled treatment group data analysis (S8 Table).

## Multivariate Cox regression and correlation analysis for survival

The survival analysis by multivariate Cox regression revealed that UFH, iNO, and PS were protective factors (S9 Table), with the four UFH-containing groups having the lowest hazard ratio after adjustment. PTX was an extremely high-risk

**Table 2. Biochemical analysis of phospholipids and proteins in bronchoalveolar lavage fluid samples.**

| Group | n | TPL | DSPC | DSPC/TPL | TP | DSPC/TP |
|---|---|---|---|---|---|---|
| | | mg/kg | mg/kg | % | mg/kg | mg/g |
| A. Treatment groups | | | | | | |
| C0 | 12 | 3.48±1.65 | 1.56±0.79 | 48.0±22.7 | 15.9±9.4 | 113.7±68.3 |
| C | 10 | 3.41±2.01§§ | 1.35±0.95§ | 38.5±15.1 | 18.8±5.5 | 68.2±34.3§§§ |
| M | 12 | 1.32±0.71 | 0.55±0.43 | 42.8±27.6 | 30.8±16.1 | 18.2±12.5 |
| MS | 11 | 3.45±1.43§§ | 1.38±0.76§ | 39.5±10.7 | 35.1±13.1^ | 42.9±25.6 |
| MH | 10 | 2.03±0.71 | 0.77±0.34 | 40.0±15.0 | 40.0±12.0^^ | 19.6±7.6^^^ |
| MSH | 11 | 2.34±0.98 | 1.07±0.44 | 48.0±21.2 | 31.3±10.6 | 34.2±11.4^^ |
| MN | 12 | 1.51±0.45^^ | 0.75±0.38 | 49.2±23.1 | 34.4±9.0^ | 23.4±12.4^^^ |
| MSN | 11 | 2.31±0.62 | 1.19±0.52 | 51.2±16.4 | 36.0±10.7^^ | 34.5±16.7^^ |
| MHN | 12 | 1.61±0.84^^ | 0.70±0.33 | 46.6±22.4 | 38.9±10.7^^^ | 19.3±9.4^^^ |
| MSHN | 13 | 2.74±1.12§ | 1.32±0.69§ | 46.3±17.7 | 35.8±9.4^^ | 36.9±21.2^^ |
| B. Pooled treatment groups | | | | | | |
| all-PS | 46 | 2.71±1.14*** | 1.24±0.61*** | 46.3±16.9 | 34.6±10.7 | 37.1±19.2*** |
| non-PS | 46 | 1.60±0.72 | 0.69±0.37 | 44.9±22.3 | 35.9±12.4 | 20.1±10.6 |
| all-UFH | 46 | 2.20±1.00 | 0.98±0.53 | 45.4±19.0 | 36.5±10.8 | 27.9±15.8 |
| non-UFH | 46 | 2.12±1.19 | 0.95±0.62 | 45.7±20.6 | 34.0±12.3 | 29.4±19.4 |
| all-iNO | 48 | 2.05±0.94 | 0.99±0.56 | 48.2±19.6 | 36.3±9.8 | 28.6±16.9 |
| non-iNO | 44 | 2.27±1.25 | 0.94±0.59 | 42.7±19.6 | 34.1±13.3 | 28.7±18.6 |

Values are mean±SD, n=10–13. For group definitions in A and B, see Table 1 legends. TPL, total phospholipids; DSPC, disaturated phosphatidylcholine; TP, total proteins. ^$p<0.05$, ^^$p<0.01$, ^^^$p<0.001$ vs. C; §$p<0.05$, §§$p<0.01$, §§§$p<0.001$ vs. M. ***$p<0.001$ vs. corresponding pooled non-designated drug group.

factor and was associated with survival time (r=−0.460, $p < 0.001$). $Cdyn_{mean}$ acted as a protective factor and was positively correlated with survival time (r=0.339, $p < 0.001$). However, due to the limited number of events, the odds ratio (OR) in the Cox regression approached zero. $LIS_{total}$ was modestly correlated with survival time (r=−0.326, $p=0.005$), but was not predictive of value by the Cox regression model (S9 Table).

### Generalized linear models

Both PS and NO exhibited significant independent main effects on the $LIS_{total}$, each contributing to a reduction in the $LIS_{total}$. Significant interactions were observed between NO and PS, as well as between NO and UFH, indicating that their combined effects were sub-additive (i.e., less than the sum of their individual effects). NO alone had the most potent reduction in $LIS_{total}$. Comparisons of marginal means revealed that the MSN or MHN group had a lower $LIS_{total}$ than that of UFH and PS alone (S10.1-S10.3 Tables in S10 Table). Main effects of PS, UFH, and NO all showed significant positive associations with Vv, while a significant negative interaction was observed between PS and UFH (S10.4-S10.6 Tables in S10 Table). PS, UFH, and NO each exhibited significant independent main effects on CV (Vv) reduction (S10.7-S10.8 Tables in S10 Table). Although each factor functions as a positive regulator of pH when administered individually, pairwise combinations demonstrate mutual antagonism. However, the co-administration of all three agents overcame this dual-agent antagonism, resulting in an elevation of pH (S10.9-S10.11 Tables in S10 Table). PS was an independent factor that lowered both $PCO_2$ and LAC (S10.12-S10.15 Tables in S10 Table). The same was true for NO on $PCO_2$ (S10.12-S10.13 Tables in S10 Table). PS was an independent factor for higher $DSPC_{BALF}$ (S10.16-S10.17 Tables in S10 Table). UFH exerted a significantly protective role with marked reduction of mortality risk (S10.18-S10.19 Tables in S10 Table). NO was an independent factor that lowered W/D levels (S10.20-S10.21 Tables in S10 Table). Although PS, UFH, and NO each independently improved

Cdyn$_{mean}$, a significantly negative interaction existed between PS and UFH when used in combination (S10.22-S10.23 Tables in S10 Table).

## Discussion

The etiology and pathophysiology of MAS are complex, but inflammatory responses and oxidative damage have been shown to be important mechanisms of MAS-related lung injury [35]. Treatment options and prognostic improvements for MAS are limited. In this study, we simulated the essential critical care in the near-term neonatal rabbit MAS [27], with a modified dosage of meconium suspension (100 mg/ml) to reduce the probability of large airway obstruction and facilitate even distribution in the alveolar compartment, thereby ensuing investigation of the mechanism of pharmacotherapeutic action of target medications. The treatment time course of MV for UFH, PS, and iNO, following our previous experience of meconium-induced acute respiratory failure, and up to 7 h observation of treatment effects in single, dual or triple combination on survival, attained a theoretical baseline for pre-clinical efficacy estimation of the pharmacotherapeutic action of heparin in MAS. The route and dosage of heparin were based on previous experimental studies in adult rabbits, where its blood activity by subcutaneous administration peaked within 1–2 h, and the half-life of absorption at the injection site was 12.9 ± 3.5 h [36]. The half-life of heparin in humans ranged from 30 minutes to 2 hours (1.5 h on average), and is directly related to the administered dose, relevant for plasma volume distribution [37,38].

Consistent with the preliminarily mixed-model analyses, the generalized linear regression models revealed that PS, UFH, and NO each significantly increased Vv and reduced CV(Vv) (S10.4-S10.8 Tables in S10 Table). Moreover, UFH was associated with a reduction in mortality risk (S10.18-S10.19 Tables in S10 Table). After controlling for other variables, UFH demonstrated a favorable pharmacodynamic response comparable to that of PS and NO. Significant interactions on LIS$_{total}$ were identified between NO×PS and NO×UFH, indicating that their combined effects were less than the sum of their individual effects (S10.2 Table in S10 Table). In line with this, comparisons of marginal means showed that the LIS$_{total}$ in the MSN group was significantly lower than in the MS group, and similarly, the LIS$_{total}$ in the MHN group was significantly lower than in the MH group (S10.3 Table in S10 Table). This suggests that despite the antagonism, combination therapy still provides an additive benefit, possibly approaching a ceiling effect where the two-drug combination nears the maximal achievable response. The regulatory effects of PS, UFH, and NO on pH were notably complex, involving a significant three-way interaction (S10.9- S10.11 Tables in S10 Table). For instance, when UFH was present (UFH=1), the pH in the [PS=1, NO=1] group was significantly higher than in the [PS=1, NO=0] group ($p$=0.043), demonstrating that the addition of NO increased pH in the presence of both PS and UFH-a clear manifestation of the three-way interaction (S10.11 Table in S10 Table). The complex higher-order interactions among these three agents warrant attention in follow-up studies but require further validation in larger-sample investigations. Given the limited sample size, fragmented sampling methodology, and discontinuous data in the present study, the interpretation of these analyses should be approached with considerable caution. Therefore, the current analysis should be regarded as preliminary and cannot serve as a definitive conclusion regarding clinical inference.

Studies on the use of heparin in neonatal ALI are limited, with one study showing a plasma half-life of 36−42 min in neonates between 25 and 36 weeks of gestation given 100 U/kg heparin intravenously within 4 h of birth [39]. Several studies have demonstrated the effectiveness of subcutaneous administration of 200−400 U/kg of UFH in treating ALI caused by lipopolysaccharide (LPS) (starting at day 6 of postnatal life) or high tidal ventilation (between 6−8 weeks of age) in a mouse model [40,41]. The dosage and route of UFH we chose were due to ineligibility for intravenous access and risk of blood loss, and difficulty in intravenous access. Therefore, we employed the safer and more convenient method of subcutaneous administration. Heparin is widely used as an anticoagulant in neonates, and there are no significant contraindications for its use unless there is a severe bleeding tendency [26,42]. We did not measure blood clotting status due to very limited blood volume. Instead, we observed survival with changes of Cdyn as lung mechanics, and heart rate (by ECG), over the whole MV period, implying appropriate gas exchange, ventilation-perfusion matching in

the lungs, as phenotype of MAS and target drug effects. There were no significant increments of mRNA expression of Syndecan-1, mild variation of eNOS and iNOS and mild increase in the expression of TF in the lung tissues of the meconium group, suggesting both intrapulmonary vascular endothelial and interstitial cells were not under stress with blood clotting function derangement, if any, due to meconium stimulation and/or heparin.

Since the early 1990s, studies have focused on surfactant inactivation and exogenous surfactant use in MAS and ALI [7,8,20,27]. Recent systematic reviews brought the relevance of surfactant in clinical use for ALI and ARDS again [1,13,14,43]. Similarly, in addition to its selective dilatation of pulmonary vasculature, iNO has also been used in very and extremely preterm infants with pulmonary hypoplasia whose intrapulmonary vasculature has an insufficient response to perfusion. The fetal lung development during GA 24–30 weeks was at canalicular to early saccular transition, their intrapulmonary capillary bed is underdeveloped, hence is likely not fully responsive to iNO. The animals in the current study were near-term and had already well-developed intra-alveolar capillary beds, as shown by electron microscopic findings (Fig 4). This is different from the fetal rabbit lungs at GA 25–27 days, corresponding to human GA 25–32 weeks [29,31]. NO has the potential for anticoagulation through the prohibition of platelet aggregation, inhibition of leukocyte adhesion and smooth muscle proliferation [44]. When the lung parenchymal injury is severe, iNO may be unevenly distributed in the lungs, which may lead to iNO ineffective [45]. Exogenous surfactant is aimed at restoring collapsed alveoli and facilitating even distribution of iNO to achieve better ventilation-perfusion matching. These mechanistic effects constituted our rationale in the study design and hypothesis using heparin with surfactant and iNO as three ancillary medications to assist MV.

Meconium is a potent stimulator of inflammatory mediators associated with the classical TLR-4/NF-κB pathway, and TNF-α, IL-6, IL-8, and IL-1β are the typical cytokines involved in the inflammatory process of ALI, also in MAS [34,46]. In this study, mRNA expression of multiple inflammatory mediators was mildly enhanced but moderately depressed after iNO, which deserves to be accredited. These phenomena were also seen for the constitutive mRNA expression of most molecules for SPs, alveolar fluid clearance, growth factors of alveolar epithelial, vascular endothelial and interstitial cells (S2, S4 Figs). This showcased the subphenotypes of the major lung tissue cells in response to the meconium-associated ALI under MV support.

The main benefit for survival was observed in the four groups treated with UFH, especially in the MSHN group. PS resulted in higher TPL, DSPC and DSPC/TP values in the BALF. In addition, the rate ratio of DSPC/TPL of MSH to MH was 1.65 in contrast to that of MS to M (S7 Table), indicating that UFH exerted a potential impact on the intrapulmonary metabolism of PS, both endogenous and exogenous. The underlying mechanism of the pharmacotherapeutic action of UFH in the treatment of experimental MAS is not clear yet. Recent studies have confirmed in a rat model of LPS-induced ALI that both UFH and LMWH reduced neutrophil aggregation and decreased lung permeability, and down-regulated the NF-κB signaling pathway, hence suppression of systemic inflammatory mediators induced by LPS [47,48].

In this study, there was a tendency for decreased mRNA expression in lung tissue inflammatory mediators with UFH, although the difference was not significant in comparison with the other drug intervention groups. Heparin improves gas exchange and reduces hyaline membrane formation during ALI in neonatal piglets induced by alveolar lavage and high PIP ventilation [49]. Consistent with the present study, the respiratory compliance and blood gases at the end of MV were improved in the UFH-treated groups. In addition, heparin has a protective effect on endothelial cells by protecting the integrity of the glycocalyx [50]. Syndecan-1 is an important component of the glycocalyx [51], and there was no marked change in the Syndecan-1 mRNA expression in each group of this study. Compared with non-UFH, the all-UFH group was able to significantly reduce the expression of TF mRNA, consistent with the anticoagulant effect of UFH. Whether heparin would modulate pulmonary arterial vascular tone is also debatable and requires in-depth investigation to verify [52–56]. The variable but modest changes in the expression of Ang-1, Ang-2, Tie-1 and Tie-2 suggest that pulmonary vascular endothelial structure remained largely intact during treatment. There was no Ang-2 mRNA overexpression, which should otherwise be regarded as a potential biomarker for impaired vascular permeability in infection and inflammation [57,58].

There are several limitations in this study. Unlike the clinical situation of MAS, there was no prepartum hypoxic stress before meconium. The well-ground meconium suspension excluded sticky, large particles that would otherwise cause large airway obstruction, hence impairing gas exchange. These differ from the pathogenic mechanisms in clinical MAS. We did not use different heparin dosage or route of administration, which may weaken the pharmacodynamic impact, but a single, empirical dose instead. Moreover, pure oxygen was used in MV to improve the overall survival, which incurs hyperoxic injury [59]. Given that anesthetics used in the 1980s–1990s adversely affected cardiopulmonary function in preterm rabbits, resulting in shorter survival times, this study employed lidocaine for anesthesia based on the following rationale: systemic anesthesia derived from maternal exposure, the need for tracheal intubation in newborn rabbits, and improved comfort during prolonged ventilation. Owing to the parallel ventilation design (supporting 12 animals simultaneously), flow partitioning may lead to progressive pressure decay during mechanical ventilation. Additionally, pressure adjustments carry an inherent risk of gas leakage due to interactions among multiple flow outlets in the plethysmograph-ventilator circuit, which maintains PIP-PEEP in individual units. Although PEEP could not be elevated to 4–5 cmH$_2$O, the model achieved relatively stable PEEP levels of 2–3 cmH$_2$O across all animals under parallel ventilation. These limitations warrant future studies to validate any additive or synergistic effects of heparin with type, dosage and route selection. The model itself has undergone various uses in decades and should be considered as a useful tool in the exploration and explanation of pathogenesis and pathophysiology of pulmonary and non-pulmonary morbidities in neonates in transition of postnatal life.

## Conclusions

In this study, we compared the efficacy of UFH, and its combined effects with surfactant and iNO in a standardized MV-treated MAS model in the near-term newborn rabbits. By significantly improved survival and alleviated lung injury, it exerted potential therapeutic benefit for lung protective ventilation, without provoking adverse effects. The overall findings should be construed as a baseline profile for understanding lung maturation and related subphenotyping of pulmonary morbidities. It should warrant careful clinical investigation to validate its efficacy in neonatal critical care.

## Supporting information

**S1 File. S1-S9 Tables.** Results of additional analyses.
(DOCX)

**S10 Table. Results of generalized linear models.**
(DOCX)

**S1 Fig. Kaplan-Meier survival curves of pooled groups with or without the designated treatment.** $^{**}p < 0.01$ vs. corresponding pooled non-designated drug group by log-rank test. For group definition, see Table 1 legends (n = 81–82).
(TIF)

**S2 Fig. The mRNA expression of proinflammatory cytokines and mediators from lung tissue.** Values are expressed as means and SD of $2^{-\Delta\Delta CT}$ of PCR measurements (n = 8–13). For group definition, see Table 1 legends. TLR, toll-like receptor; MyD88, myeloid differentiation primary response protein 88; NF-κB, nuclear transcript factor kappa B; TNF-α, tumor necrosis factor α; IL, interleukin.
(TIF)

**S3 Fig. The mRNA expression of endothelial cell proliferation, injury, and coagulation-related factors from lung tissue.** Values are expressed as means and SD of $2^{-\Delta\Delta CT}$ of PCR measurements (n = 6–13). For group definition, see Table 1 legends. Ang, angiopoietin; TF, tissue factor.
(TIF)

**S4 Fig. The mRNA expression of other molecules from lung tissue.** Values are expressed as means and SD of $2^{-\Delta\Delta CT}$ of PCR measurements (n = 8−13). For group definition, see Table 1 legends. SP, surfactant protein; IGF, insulin-like growth factor; VEGF, vascular endothelial GF; KGF, keratinocyte GF; AQP-5, aquaporin-5; eNOS, endothelial nitric oxide synthase; iNOS, inducible nitric oxide synthase. §$p < 0.05$ vs. M.
(TIF)

**S1 Dataset. The raw data for this study.**
(XLSX)

## Acknowledgments

We are very grateful to the parents of newborn babies for donating meconium. The technical support of Ni Qin and Jiangang Lao during the experiments is greatly appreciated. Dr. Yaling Xu, Department of Neonatology, Guangzhou Women and Children's Medical Center, is highly appreciated for advice in statistical analysis with generalized linear regression modelling of outcome estimation.

## Author contributions

**Conceptualization:** Qiang Gu, Bo Sun.

**Data curation:** Siyu Xie, Qiang Gu, Guiyin Zhuang, Xiaojing Guo, Bo Sun.

**Formal analysis:** Siyu Xie.

**Funding acquisition:** Qiang Gu.

**Investigation:** Siyu Xie, Qiang Gu, Guiyin Zhuang, Xiaojing Guo, Bo Sun.

**Methodology:** Siyu Xie, Guiyin Zhuang, Xiaojing Guo.

**Project administration:** Qiang Gu.

**Supervision:** Bo Sun.

**Writing – original draft:** Siyu Xie, Qiang Gu, Guiyin Zhuang, Xiaojing Guo, Bo Sun.

**Writing – review & editing:** Qiang Gu, Bo Sun.

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
