## [Decision Letter · Decision Letter 0]

1 Sep 2025

Dear Dr. Sun,

Thank you for submitting your manuscript to PLOS ONE. After careful consideration, we feel that it has merit but does not fully meet PLOS ONE’s publication criteria as it currently stands. Therefore, we invite you to submit a revised version of the manuscript that addresses the points raised during the review process.

We look forward to receiving your revised manuscript.

Kind regards,

Elsayed Abdelkreem, MD, PhD

Academic Editor

PLOS ONE

Journal Requirements:

2. We note that your Data Availability Statement is currently as follows: All relevant data are within the manuscript and in Supporting Information files.

Reviewers' comments:

Reviewer's Responses to Questions

**Comments to the Author**

1. Is the manuscript technically sound, and do the data support the conclusions?

Reviewer #1: Partly

Reviewer #2: Partly

2. Has the statistical analysis been performed appropriately and rigorously?

Reviewer #1: Yes

Reviewer #2: No

3. Have the authors made all data underlying the findings in their manuscript fully available?

Reviewer #1: Yes

Reviewer #2: Yes

4. Is the manuscript presented in an intelligible fashion and written in standard English?

Reviewer #1: Yes

Reviewer #2: No

Reviewer #1: In this study, Xie et al investigate the effects of heparin in near term newborn rabbits in a model of mechanical and chemical induced lung injury in an effort to identify therapeutic options for meconium aspiration syndrome. I commend the authors on using appropriate controls for the various experiments. I have the following concerns:

-One important issue with heparin is that is also known to bind to angiogenic factors and can inhibit late stage alveolar development. There are several studies that demonstrate that heparin inhibits lung growth and can potentiate lung injury as a result of this mechanism. The authors should consider this additional data especially in light of their own findings. This concern should be addressed in a revised version of the manuscript. For example angiogenic factors such as VEGF and VEGFR activation can be evaluated in lung samples.

-I do think that it is essential to consider measuring coagulation parameters in the groups that received heparin vs. not (e.g. Factor Xa assay vs. partial thromboplastin time measurement)

-An additional concern is the effect of lung injury appears to be minimal based on the lung injury scoring (Table 2). Is this similar to other models/expected? Can the authors comment? My concern would be is that the disease is not severe enough and therefore we cannot make any conclusions on whether or not heparin itself has an additive effect on MAS treatment or we are just seeing the well established effects of surfactant/iNO?

Reviewer #2: Thank you for the opportunity to review the manuscript entitled “Efficacy of heparin in respiratory support of near-term newborn rabbits with meconium-induced acute lung injury.” This study investigates whether adding heparin to standard therapy improves outcomes in a meconium aspiration-induced lung injury model. The topic is interesting and clinically relevant; however, there are several critical issues that need to be addressed. In particular, the unsophisticated statistical approach—comparing multiple parameters across many groups—makes interpretation difficult. Furthermore, some results raise concerns about the validity of the acute lung injury model itself. My main concerns are as follows:

1. Language quality – The manuscript contains numerous grammatical errors and unclear expressions. Language editing by a native English speaker is strongly recommended.

2. Anesthesia and lidocaine use – Please clarify how the pups were anesthetized. Why was lidocaine added to the supplemental fluid? Also, it is unclear whether the administered dose was appropriate.

3. Route of heparin administration – The cited ARDS reference showing potential benefit of heparin used nebulized administration. Please justify why subcutaneous administration was chosen for this study.

4. Validity of the lung injury model – The meconium aspiration group (M group) did not show a higher wet/dry ratio than the control group (C group), raising concerns about whether this model accurately reflects acute lung injury.

5. Statistical approach – Given the large number of groups and outcome measures, the results are difficult to interpret. Since the primary aim is to determine the effect of adding heparin to standard therapy, the analysis should focus on this aim. Using generalized linear models to assess the independent and interactive effects of each treatment would be more appropriate.

6. Data presentation – Including figures of selected key parameters (e.g., biochemical markers, qPCR results) would help readers better interpret the findings.

**Do you want your identity to be public for this peer review?** For information about this choice, including consent withdrawal, please see our Privacy Policy

Reviewer #1: No

Reviewer #2: No

---

## [Author Response · Author response to Decision Letter 1]

15 Oct 2025

Response to Reviewers' 20251015

We are very grateful to the reviewers for their comments regarding the improvement of our manuscript. We have made thorough revision according to the points raised by both reviewers and academic editor, and summarized response point-by-point below, to the queries (Qx) in order of the decision letter of 20250901. In the current version (R1), yellow highlights indicate new or modified content, while red words/fonts denote deleted content.

Review Comments to the Author

Reviewer #1:

Q1. In this study, Xie et al investigate the effects of heparin in near term newborn rabbits in a model of mechanical and chemical induced lung injury in an effort to identify therapeutic options for meconium aspiration syndrome. I commend the authors on using appropriate controls for the various experiments.

Response:

Thanks for the comments. Regarding the control groups, our preliminary design was to evaluate, in an established near-term newborn rabbit MAS model, the efficacy of heparin, either standing alone or in combination with surfactant and/or inhaled NO. The controls were in two groups, either not receiving meconium (group C), or given meconium but no additional specific medications (group M), to undergo standardized tidal volume ventilation and to set best and worst survival limit, as shown by Kaplan-Meier survival curves (Fig. 2). This enabled evaluation of the therapeutic action of specific respiratory medications, hence explore the underlying mechanisms. There was a technical limitation for the groups related to exposure to iNO as there was no additional multi-plethysmograph-ventilator to ensure intra-litter randomization. Therefore, in the second phase of the experiment, groups were all exposed to iNO through a mixed inter- and intra-litter randomization, in which heparin and PS were randomized within litters, thus comparable to those in the non-iNO groups in randomization. In both phases, the ventilation settings and other medications were similar following the experimental protocols. In the marked R1, on page 14, lines 289–292, we have modified the description of statistic tests for the groups in two subsets: one by using group M and C as controls, and three groups without NO (MS, MH, MSH); and the other, the four groups with NO (MN, MSN, MHN, MSHN) were compared separately versus the group C and M. One-way ANOVA (F test) was used with post hoc test for repeated measures. There was no direct comparison of the two corresponding treatment groups across the two sets (i.e., MH vs. MHN, MS vs. MSN, and MSH vs. MSHN, respectively). As the mean values, SD and sample size are available, readers should be able to readily estimate the differences in-between.

Surfactant is off-label used in daily practice in NICU for late preterm and term/post-term newborn infants at risk of respiratory distress due to meconium aspiration, while iNO is applied for persistent hypoxemic respiratory failure complicated with persistent pulmonary hypertension (PPHN).

I have the following concerns:

Q2. - One important issue with heparin is that is also known to bind to angiogenic factors and can inhibit late stage alveolar development. There are several studies that demonstrate that heparin inhibits lung growth and can potentiate lung injury as a result of this mechanism. The authors should consider this additional data especially in light of their own findings. This concern should be addressed in a revised version of the manuscript. For example angiogenic factors such as VEGF and VEGFR activation can be evaluated in lung samples.

Response:

Thanks for the concern about long-lasting adverse effects of heparin on the angiogenesis of neonatal lungs. As proposed, in the marked R1, on page 5, lines 101–109, we read the reference (ref. 23 in R1) about heparin hindering late alveolar development in a mature mouse model. Their dosage of heparin was several times higher than what we provided, and their conclusions may not be generalizable considering our peculiar experimental design and overall and specific efficacies, and possible half-life effect of heparin. We also performed additional examination by PCR for lung tissue mRNA expression of Ang-1, Ang-2, and their receptors, Tie-1 and Tie-2, from the lung tissue samples of all the treatment groups. The data are included in supplemental information S3 Fig. and S8 Table of R1. There is no evidence to consider that the intrapulmonary vasculature was substantially impaired attributable to heparin and associated other medications.

The role of Ang-2 in angiogenesis remains unclear. It primarily exerts anti-angiogenic effects by inhibiting Ang-1-induced Tie-2 phosphorylation, while Tie-1 blocks the Ang-1 signaling pathway by inhibiting Tie-2. In an in vitro study, it indicates that in the absence of Ang-1, Ang-2 can act as a weak agonist for Tie-2, promoting phosphorylation of Tie-2, thereby inhibiting endothelial cell apoptosis and enhancing migration. (Yuan HT, Khankin EV et al. Mol Cell Biol. 2009, 29(8): 2011-22). Tie-1 and Tie-2 exist in hetero-oligomeric complexes on the endothelial surface, with Tie-1 inhibiting Tie-2. When VEGF levels transiently increase, it stimulates Tie-1 cleavage, thereby enhancing Ang-1 signaling. Furthermore, Ang-2 overexpression is associated with increase in vascular permeability and may be considered an inflammation-associated marker [ref. 56, 57 in R1]. Notably, mRNA expression of these molecules and mediators in this study may undergo a developmental change in magnitude as response to the intervention in early postnatal life of the experimental animals. For technical difficulty (lack of antibody against specific rabbit antigen), we did not measure their metabolites of mRNA expression (by ELISA or western blot), nor for their time course alteration. As mentioned above, it also requires specific study plan and protocol to verify.

Q3. - I do think that it is essential to consider measuring coagulation parameters in the groups that received heparin vs. not (e.g. Factor Xa assay vs. partial thromboplastin time measurement)

Response:

Thanks for this recommendation. In the original manuscript, we have reported mRNA expression of Syndecan-1, tissue factor (TF) and VEGF to connect the intrapulmonary vascular status, in part associated with coagulation as explained in the above response to Q2. Regarding the pharmacological action of anticoagulation by heparin, due to the very low blood volume accessible at the end of experiment for tiny pups (BW circ. 40-55 g), and to share it with other measurements (blood gas), it was not considered to measure blood clotting and coagulation by APTT or ACT. We examined hemorrhage in the lung injury score (LIS, items and total, Table 1 and S5 Table in R1) measurement. There was no tangible evidence of excessive bleeding in the lungs from those exposed to heparin. In the marked R1, on page 6, lines 112–118, we have added a reference for heparin use in neonates by the American Pharmacists Association (ref. 26 of R1), in which there was no prohibition or restriction in neonatal infants unless its use in ECMO, in which it should be subject to repeated measures of ACT for systemic heparinization. Considering topic use of heparin in indwelling catheterization at NICU practice, the adverse effects of heparin may be confined to routine clinical pharmacology surveillance. Studies evaluating the risk of UFH-related bleeding in the neonatal and pediatric populations are limited. A well-design experiment using relatively larger animal model should be expected to test the coagulation of heparin in this kind of MAS-ALI model to answer the query. And prospective studies are needed to determine the precise risk of UFH-associated bleeding. Nevertheless, the common cause of fatal UFH-induced bleeding is due to accidental overdose [ref. 41 of R1].

Q4. - An additional concern is the effect of lung injury appears to be minimal based on the lung injury scoring (Table 2). Is this similar to other models/expected? Can the authors comment? My concern would be is that the disease is not severe enough and therefore we cannot make any conclusions on whether or not heparin itself has an additive effect on MAS treatment or we are just seeing the well established effects of surfactant/iNO?

Response:

Thanks for this comment. We do not consider the lung injury severity should be minimal. The measurements of lung injury type and severity should consist of LIS and prevalence of pneumothorax (PTX). Other variables like Cdyn, W/D, DSPC/TP may add to understanding the injury pattern, magnitude, and underlying mechanisms for lung liquid clearance, surface tension vs. vascular-to-alveolar permeability, respectively. The meconium preparation, administration, and response pattern in current experiment are similar to our previous ones (Xu Y et al, Pediatr Res 2023, ref. 27 in R1). The major difference was the doubled concentration of meconium in suspension (100 mg/ml, 4 ml/kg) in current study so as to minimize fluid load, which was associated with the lower early deaths and PTX compared to the previous one [final survival rate 67% (14/21) vs. 26.5% (16/61); 10% (2/21) vs. 23% (14/61), respectively], along with reduced inflammatory response. This improvement may be attributed to the reduced viscosity of meconium suspension, which minimized large airway obstruction but retained underlying hypoxia-induced, hyperoxic stress-mediated, inflammatory response. [As a note, the previous MAS study (ref. 27) required additional meconium dose (1-3 mL/kg) in part of the animals in each group to achieve hypoxemic respiratory failure and lung injury before the commence of surfactant treatment. Thus, it incurs large fluid volume load to the lung mechanics, injury severity and overall survival differences]. The difference in current survival curve of all treated groups may also be related to more delicate management during ventilation, and the type of surfactant preparations [in current study as BALF-derived, containing higher phosphatidylcholine (PC) vs. lung tissue homogenate-derived, containing high phosphatidylethanolamine (PE) of non-type II cells, or synthetic peptide-phospholipids of selected phosphatidylcholine (PC) and phosphatidylglycerol (PG) in ref. 27] as well as the use of iNO and heparin.

Human meconium is bacteria-free, contains neutral lipids, and may not elicit infection. The pathophysiology of MAS is associated with systemic hypoxemia with lung exposed to hyperoxic stress. Therefore, the lung injury should be considered non-infectious. We relied on the measurement of lung mechanics, lung histopathology, morphometrics and LIS, surfactant phospholipid pools, and expression of mRNA in lung tissues representative of different epithelial and endothelial cell types, to characterize the lung injury type and severity, and the efficacy of heparin and its combination with PS and iNO. Different from preterm lungs, which are at late canalicular to early saccular stage of development, and prone to respiratory distress syndrome (RDS), the near-term rabbit lungs are at alveolar stage, with plenty of surfactant phospholipids and specific proteins (SP-A, B, C). Thus, the lung injury pattern should be characterized as in term neonatal lungs with sufficient surfactant pools, which are >10 times at birth that of the adult and children, with meconium-induced chemical alveolitis, partial airway obstruction, and impaired endogenous surfactant metabolism. This point is addressed in the Results and Discussion of R1.

In the marked R1, on page 12, lines 243–247, we have re-examined all the lung tissue sections for histopathology and performed LIS again by including meconium distribution in alveolar space and small airways as an additional item in the lung injury score (LIS). This is added in supplemental table S5 and Table 1 (in both marked and clean body text content of R1), and should more comprehensively account for the lung injury pattern. Based on the survival length and rate, lung mechanics, LIS, lung surfactant phospholipid pools, and integrated mRNA expression levels of relevant molecules of lung cells, the impact of heparin was discernible with PS and iNO as shown in S2-S9 Table.

Reviewer #2:

Thank you for the opportunity to review the manuscript entitled “Efficacy of heparin in respiratory support of near-term newborn rabbits with meconium-induced acute lung injury.” This study investigates whether adding heparin to standard therapy improves outcomes in a meconium aspiration-induced lung injury model.

The topic is interesting and clinically relevant; however, there are several critical issues that need to be addressed. In particular, the unsophisticated statistical approach—comparing multiple parameters across many groups—makes interpretation difficult. Furthermore, some results raise concerns about the validity of the acute lung injury model itself. My main concerns are as follows:

Q5. 1. Language quality – The manuscript contains numerous grammatical errors and unclear expressions. Language editing by a native English speaker is strongly recommended.

Response:

Thanks for the comments. We have extensively revised the manuscript using tools for spelling, writing and grammar. Long sentences are shortened, data presentation, interpretation, concept and points are expressed more succinctly, for comprehension. All the changes are marked in R1.

Q6. 2. Anesthesia and lidocaine use – Please clarify how the pups were anesthetized. Why was lidocaine added to the supplemental fluid? Also, it is unclear whether the administered dose was appropriate.

Response:

Thanks for the comments. The analgetic used for does in sequence of diazepam and urethane, along with a face mask and oxygen were to minimize the risk of hypoxemia-associated fetal stress during delivery. The newborn pups were taken out by C-section, dried and weighed. Initial anesthesia was administered via intraperitoneal injection of diluted lidocaine solution to facilitate intratracheal intubation. Subsequent lidocaine was given intraperitoneally with glucose and sodium bicarbonate diluted in different proportions for respective physiological need and relevance (lines 161-162 and 174-176, page 8, in marked R1). This initial anaesthetic procedure and protocol was adopted from the early experiments, in the 1980-1990 era, of short-time (30 min-5 hours) ventilation of rabbits at 27-30 days of gestational age (GA, term 31 days) at Karolinska experimental perinatal pathology laboratory (Dr. Bengt Robertson), Stockholm, Sweden. For peritoneal intermittent injection, it also followed our previous experience with ventilated very preterm rabbit pups (GA 26 days) studies, which provided glucose, sodium bicarbonate and lidocaine for energy, counter balance acidosis, and sedation, respectively, and ensured prolonged survival up to 22 hours (ref. 29 in R1). No adverse events were identified directly from the specified medication.

Q7. 3. Route of heparin administration – The cited ARDS reference showing potential benefit of heparin used nebulized administration. Please justify why subcutaneous administration was chosen for this study.

Response:

Thanks for the comments. The subcutaneous heparin was derived from the reference in adult animal model (ref. 35 in R1). The dosage was based on the most investigational reports for adults and children, following the recommendation from American Pharmacist Association year book (ref. 26 in R1). Due to its half-life in several hours, and being impossible for continuous intravenous injection, a single dose of 100 U/kg BW should be relevant for observation and comparison of its efficacy in parallel or in combination with intratracheal PS and iNO. The measurements of pharmacotherapeutic action are preliminary for understanding its role in the specified neonatal pathological model for clinical implications and construed phenotypes. Due to very low bioavailability, it is not worth delivering heparin alone, or with surfactant as carrier, through aerosol to these tiny pup lungs.

In neonates with central venous access devices (CVADs), a continuous infusion of UFH at 0.5 U/kg/h is recommended to maintain CVAD patency. For umbilical artery catheters (UAC), prophylactic anticoagulation with UFH at a

---

## [Decision Letter · Decision Letter 1]

1 Dec 2025

Dear Dr. Sun,

Thank you for submitting your manuscript to PLOS ONE. After careful consideration, we feel that it has merit but does not fully meet PLOS ONE’s publication criteria as it currently stands. Therefore, we invite you to submit a revised version of the manuscript that addresses the points raised during the review process.

We look forward to receiving your revised manuscript.

Kind regards,

Elsayed Abdelkreem, MD, PhD

Academic Editor

PLOS ONE

Journal Requirements:

Reviewers' comments:

Reviewer's Responses to Questions

**Comments to the Author**

Reviewer #1: All comments have been addressed

Reviewer #2: (No Response)

2. Is the manuscript technically sound, and do the data support the conclusions?

Reviewer #1: Yes

Reviewer #2: Partly

3. Has the statistical analysis been performed appropriately and rigorously?

Reviewer #1: Yes

Reviewer #2: No

4. Have the authors made all data underlying the findings in their manuscript fully available?

Reviewer #1: Yes

Reviewer #2: Yes

5. Is the manuscript presented in an intelligible fashion and written in standard English?

Reviewer #1: No

Reviewer #2: No

Reviewer #1: The authors have addressed my comments and this manuscript is acceptable for publication in PLOSOne in its current format.

Reviewer #2: Thank you for the opportunity to review the revised version of manuscript entitled “Efficacy of heparin in respiratory support of near-term rabbits with meconium-induced acute lung injury”.

The manuscript has improved following revision; however, the statistical analysis remains inappropriate, and the data presentation is still difficult to follow. As noted below, the data should be analyzed using a generalized linear model to assess the effects of each treatment (UFH, pulmonary surfactant, and NO inhalation) as well as their interactions. Furthermore, this study appears to aim at identifying optimal treatment combinations rather than solely evaluating the efficacy of UFH. Please consider revising the entire manuscript, including the title, to clearly reflect this objective.

The details of concerns are as follows:

Major

1. Intravenous lidocaine alone cannot provide general anesthesia. It appears that no other general anesthetic agents were administered to the neonatal rabbits. Please justify why lidocaine-only anesthesia was considered appropriate.

2. Why were the animals ventilated with 2–3 cmH₂O PEEP? This level seems relatively low and may predispose to lung collapse. Please explain the rationale for selecting these PEEP settings.

3. Although the information on group allocation has improved, the statistical analysis remains inadequate. This study evaluates the efficacy of three interventions (UFH, pulmonary surfactant, and NO inhalation) and their interactions in an MAS model. A generalized linear model would be more appropriate. Please consider reanalyzing all data using generalized linear analysis. In addition, the approach to variable selection in the multivariate Cox regression model appears inappropriate. The analysis should assess the effect of each treatment and their interactions.

4. Please include PaO₂ data from the blood gas analysis.

Minor

1. Were all animals administered meconium from the same batch? Please clarify this point.

**Do you want your identity to be public for this peer review?** For information about this choice, including consent withdrawal, please see our Privacy Policy

Reviewer #1: No

Reviewer #2: **Yes:** Kentaro Tojo

---

## [Author Response · Author response to Decision Letter 2]

15 Jan 2026

Response to Reviewers' comments 20260115

We are very grateful to the reviewers for their further comments regarding the improvement of our manuscript (R1). We have made thorough revision according to the points raised by one of the reviewers, mainly regarding the “generalized linear model”, to characterize the sole and interactive effects of the three medications, and other technical issues in the experiment. We have summarized the response point-by-point below, to the queries (Qx) in order of the decision letter of 202512. In the current version (R2), yellow highlights indicate new or modified content, while red words/fonts denote deleted content referring to the context as marked version.

Review Comments to the Author

Reviewer #2:

Thank you for the opportunity to review the revised version of manuscript entitled “Efficacy of heparin in respiratory support of near-term rabbits with meconium-induced acute lung injury”.

Q1. The manuscript has improved following revision; however, the statistical analysis remains inappropriate, and the data presentation is still difficult to follow. As noted below, the data should be analyzed using a generalized linear model to assess the effects of each treatment (UFH, pulmonary surfactant, and NO inhalation) as well as their interactions. Furthermore, this study appears to aim at identifying optimal treatment combinations rather than solely evaluating the efficacy of UFH. Please consider revising the entire manuscript, including the title, to clearly reflect this objective.

Response: Thanks for the comments. We analyzed part of the data using generalized linear models, as detailed in below response to question 4 (Q4).

In the current revision (R2), we have conducted a generalized linear regression analysis with survival status as the primary outcome (Y) in the model as a dependent variable [f (Y)]. By incorporating all main effects of PS, UFH, and NO as well as all possible two-way and three-way interaction terms (i.e., a full factorial model), we stepwise characterized their respective effects. The analysis used other variables derived at the end of survival, with variable ventilation length, LIStotal, Vv, CV (Vv), DSPCBALF, pH, PCO2, LAC (in heart blood), W/D, Cdynmean (derived from the last 7 hours during the 10-h total ventilation length) instead of survival length. Our major findings suggest that:

1) Both PS and NO exerted significant independent main effects on LIStotal, each with an overt reduction of LIStotal. However, significantly negative interactions were observed between NO×PS and NO×UFH, indicating that their combined effects were less than the sum of their individual effects. NO alone exerted the strongest reduction in LIStotal. Based on comparisons of marginal means, the concurrent use of PS and NO (MSN group) or UFH and NO (MHN group) resulted in a lower LIStotal than the use of UFH and PS alone (S10.1- S10.3 Table).

2) Main effects of PS, UFH, and NO all showed significant positive associations with Vv, while a significant negative interaction was observed between PS and UFH, indicating that their combined use resulted in a smaller increase in Vv than the sum of their individual effects (S10.4- S10.6 Table).

3) The main effects of PS, UFH, and NO were each significantly associated with a decrease in CV (Vv). This indicates that the effects of these three factors on CV(Vv) are primarily independent and additive (S10.7- S10.8 Table).

4) While each factor was a positive regulator of pH as a single outcome item, pairwise combination comparisons showed mutual antagonism. The concomitant administration of all three, however, revealed a synergistic/compensatory mechanism that partly overcame this interactive antagonistic mechanism, thereby resulting in significantly elevated pH values (S10.9- S10.11 Table).

5) PS and NO were independent factors that lowered PCO₂ (S10.12-S10.13 Table). PS was an independent factor that lowered LAC (S10.14-S10.15 Table). PS was an independent factor for higher DSPCBALF (S10.16-S10.17 Table).

6) UFH was a significant protective factor, associated with a significantly reduced risk of mortality (S10.18-S10.19 Table).

7) Only NO exerted an independent main effect that significantly lowered W/D levels (S10.20-S10.21 Table).

8) Both PS and UFH, as well as NO, can independently improve Cdynmean. However, a significantly negative interaction existed between PS and UFH when used in combination (S10.22-S10.23 Table).

We summarized above finding in the section of Methods/ statistical analysis (lines 296 to 303 on page 14 of R2), Results/Regression models (lines 489 to 508 on page 24 of R2), and Discussion/implications (lines 528 to 549 on page 25 of R2) of survival analysis using the generalized linear regression models, and amended the Cox HR regression for survivals (lines 456 to 459 on page 21 of R2). To facilitate reviewers' evaluation, a separate file entitled "S10.1-S10.24 Tables," which details each generalized linear regression model, is attached as supplemental material with this revision (R2). As a result, we have modified the article title of R2 to be more informative, highlighting the “generalized linear regression model” in the estimation of efficacy.

There is a paucity of other medications in addition to the PS and/or iNO regimens in the literature. Likewise, UFH therapy for MAS remains underexplored in existing literature, though it was reported in pediatric and adult respiratory and critical care, including preclinical experimental pharmacology and pathophysiology of ALI (refs. 41, 47, 49 in R2). Therefore, this study focused on evaluating the pharmacodynamic response of UFH in MAS, with particular attention to potentially synergistic interactions with PS and/or NO. In the Results and Discussion sections of R1, we only addressed the synergistic effects from the phospholipid components in BALF and LH compartments. We did find the impact of UFH on the phospholipid metabolic pools (lines 433 to 438 on page 20 and lines 603 to 606 on page 28 of R2). Specifically, our findings demonstrate that UFH and PS exhibit a synergistic effect, and similarly, UFH and iNO also act synergetically.

We modified and re-analyzed the Cox hazard ratio (HR) survival regression models, while BW, Cdynmean, and LIStotal were valid parameters to be included; therefore, this table should be retained in R2, supplemental S9 Table, for comprehension.

Q2. Intravenous lidocaine alone cannot provide general anesthesia. It appears that no other general anesthetic agents were administered to the neonatal rabbits. Please justify why lidocaine-only anesthesia was considered appropriate.

Response: Thanks for the comments.

The does received sedatives (diazepam) and anesthetics (urethane, repeated dosage) should have impacted the fetuses before delivery (ref. Meissner J, Preil P. Distribution of 14C-Diazepam (Valium) in pregnant and lactating rabbits. Nucl Med (Stuttg). 1975 Aug 31;14(3):272-84; Kanto JH. Use of benzodiazepines during pregnancy, labour and lactation, with particular reference to pharmacokinetic considerations. Drugs. 1982 May;23(5):354-80; Field KJ, Lang CM. Hazards of urethane (ethyl carbamate): a review of the literature. Lab Anim. 1988 Jul;22(3):255-62).

In the 1980s-1990s, a combination of acepromazine and ketamine or sodium pentobarbital and pancuronium bromide were used for general anesthesia of neonatal rabbits (ref. 7, 8, 20, 32 in R2). These regimens were employed in short-term (30 minutes to 5 hours) ventilation studies in newborn rabbits at 27-30 days of gestation (full-term gestation: 31 days). However, due to the immature metabolism/catabolism in immature rabbits, these anesthetic protocols frequently caused respiratory depression, making them unsuitable for long-term survival observations. In the latter 1980s, investigators tended to apply a mild anaesthesia without additional i.p. anaesthesia, but pacuronium in some, in the newborn rabbits subjected to the experiment with short-time (60 min) mechanical ventilation (Fiscone JM, Jacobs HC, et al. Pediatr Res 1987; 22(6): 730-735; Fiascone JM et al. Exp Lung Res 1989; 16: 311-321; Mercury MR et al. J Appl Physiol 1989; 66(5): 2039-2044).

Based on our previous experience with ventilation studies in newborn rabbits (gestational age 26-30 days), does received intramuscular injections of diazepam and urethane for sedation and anesthesia, followed by intravenous supplementation of urethane to maintain the doe anesthesia till completion of delivery. Like other anesthetics for general anesthesia, diazepam and urethane can cross the placental barrier within a short period, resulting in newborn pups being born in a semi-anesthetized state. Immediately after birth, initial anesthesia was achieved via intraperitoneal injection of diluted lidocaine solution, supplemented by subcutaneous infiltration of diluted lidocaine for local anesthesia around the front neck during tracheal intubation to the pups when necessary. Throughout the ventilation period, a mixed solution containing 2% lidocaine, 5% NaHCO₃, and 10% glucose (vol/vol/vol 1:3:6 or 2:3:5) was administered every 1-2 hours to each pup. This combination provided triple pharmaceutical actions: glucose for energy supply, NaHCO₃ for correction of metabolic acidosis, and lidocaine for anesthesia.

The protocol for mechanically ventilated newborn rabbits was construed to simulate clinically critical care with minimum stimulation and avoid drug adverse effects. Thus, it exerts mild anesthetic effects while the spontaneous respiratory effort shown as chest and abdominal movement during mechanical ventilation was maximally preserved, for prolonged survival (ref. 27-31 in R2). During mechanical ventilation, for those with good response, spontaneous breath movement, with no distress-induced body movements were observed, along with stable heart rates (150-250 beats/min) throughout (unless early death occurred). Direct adverse events attributable to these pharmacological agents were closely observed (pneumothorax, ECG tracings) (ref. 27-31 in R2). For those with poor response from the initial 15-60 min of ventilation, they tended to be pale or cyanotic in the head, body and limbs, lower heart rate alteration (50-100/min), with ECG S-T fragment alteration denoting cardiac ischemia and/or atrioventricular block. They may survive shortly with transient improvement in oxygenation. These were found mainly in those who died within the initial 1-2 hours.

From one of the coauthors (B.S)’s experience in the 80s and 90’s when both pentobarbital and pancuronium were used, the preterm animals (GA27d) may survive for only 30-60 min, which was the time limit in most of the studies then using preterm rabbits at GA 27-27.5 days, with or without ECG monitoring. Their ventilation was often with very high tidal volume (10-12 ml/kg) resulting in a very fast heart rate (250-300/min, being tachycardia), which was not physiologically acceptable from current knowledge of lung protective ventilation strategy in neonatal critical care. Likewise, in the studies with pups of GA in 29/29.5/30 days, pentobarbital and pancuronium may also cause adverse effects in cardiopulmonary function during mechanical ventilation. Thus, the analgesic use with lidocaine in our experiments was based empirically on the maternal general anesthesia to fetuses, tracheostomy at birth as resuscitation to avoid hypoxic fatality, and comfort during prolonged ventilation, which seemed amenable within the modified study protocol. This technical limitation is now added in the discussion-limitation, lines 636-640 on page 29 of the R2 marked version. A short phrase is added as “subcutaneous infiltration of the anterior neck with the same lidocaine, when needed” in R2-marked version, Materials and methods, Animal management and MAS model (lines 148-151 on page 7).

Q3. Why were the animals ventilated with 2–3 cmH₂O PEEP? This level seems relatively low and may predispose to lung collapse. Please explain the rationale for selecting these PEEP settings.

Response: Thanks for the comments.

Since the 1970s and 1980s, investigators have initiated mechanical ventilation in preterm and near-term rabbits at GA 27-30d. They observed that epithelial cell damage, associated with alveolar structure immaturity, surfactant deficiency, and uneven alveolar expansion collectively contributed to acute lung injury, which constituted the underlying pulmonary pathology of respiratory distress syndrome (RDS) in the preterm rabbits (Nilsson R, Grossmann G, Robertson B. Pediat Res 1978; 12:249-255; Nilsson R. Acta Anaesthesiol Scand. 1982 Apr;26(2):89-103). At that time, the equipment was already capable of providing mechanical ventilation to multiple animals in parallel for testing surfactant in the improvement of lung mechanics, i.e., dynamic compliance of respiratory system (Cdyn, or Crs). In those studies, PEEP was not used to avoid masking the surfactant effects. Moreover, the pressure entering each animal lungs could not be individually adjusted (Lachmann B. Pediatr Res 1981; 15:833-838; 1982; 16:921-927). In the meantime, others reported individually adjusted PIP-PEEP based on “one ventilator-to-one pup” settings (Tooley W, Clements JA, et al. Am Rev Respir Dis 1987; 136:651-656), which set ventilator PEEP only 1 cmH2O with very fast frequency (48 beat/min) to achieve (supposedly with intrinsic) PEEP at 4 cmH2O, and achieved very high tidal volume, up to 10-15 ml/kg within 60 min.

Lately, one of the co-authors (BS) working at Karolinska experimental perinatal pathology laboratory (Head, Dr. Bengt Robertson), modified the equipment to enable pressure regulation for each individual animal, achieving a ventilation mode with target tidal volume (Vt) with PIP-PEEP. This emphasized the importance of animal models that conform to respiratory physiology for comparing the effects of different surfactant preparations. (Sun B, Kobayashi T, Cursted T, Grossmann G, Robertson B. Eur Respir J. 1991;4:364-70).

Subsequently, it revealed that appropriate PEEP (2-3 cmH₂O) promotes alveolar recruitment in GA27d preterm rabbits without increasing the incidence of pneumothorax (was up to 50%), thereby facilitating the establishment of mechanical ventilation models with PIP-PEEP restricted tidal volume below 9 ml/kg (Rider ED, Jobe AH, Ikegami M, Sun B. J Appl Physiol 1992; 73(5): 2089-2896). This was in contrast to their previous studies with no PEEP but very large tidal volume of 10-15 ml/kg by individually adjusted PIP in testing efficacy of perinatal medications including surfactant preparations in preterm rabbits (Ikegami M, Jobe AH, et al. J Clin Invest 1987; 79:1371-1378; Am Rev Respir Dis 1987; 136:892-898).

The multi-channel mechanical ventilation model underwent further refinement, enabling a series of studies on the pathogenesis and intervention efficacy of diseases associated with preterm-to-near-term rabbits with bacterial pneumonia, or meconium aspiration (Sun B, Robertson B et al. Biol Neonat 1993; 63:96-104; Herting E, Sun B, Robertson B, et al. Pediatr Res 1994; 36:784-791; ref. 30 in R2).

Previously reported data from our team indicate that PEEP modifies the response of preterm rabbits to surfactant during ventilation (ref. 30 in R2). In recent years, our research group has successfully established a multi-channel parallel mechanical ventilation model in newborn rabbits at gestational days 26–30, simulating human lung development in third trimester: across the late canalicular, early and late saccular, and early alveolar stage before term birth, and alveolar stage at early postnatal life (Zhuang G et al, Exp Biol Med 2026 accepted). These were achieved by optimizing Vt at 4-6 mL/kg and PEEP at 2–3 cmH₂O (ref. 27, 28, 30 in R2). Given the parallel ventilation design of our model (supporting up to 12 animals simultaneously), the flow partitioning effect may lead to progressive pressure decay during mechanical ventilation. Additionally, pressure adjustments carry inherent risks of air leakage due to interaction among the gas flow outlets for maintenance of PIP-PEEP level in gradient in individual units, within the plethysmograph-ventilator circuit that was provided with a very high main gas flow from the ventilator. While we may not achieve PE

---

## [Decision Letter · Decision Letter 2]

10 Mar 2026

Efficacy of heparin in respiratory support of near-term rabbits with meconium-induced acute lung injury. Linear regression model analyses

PONE-D-25-35179R2

Dear Dr. Sun,

We’re pleased to inform you that your manuscript has been judged scientifically suitable for publication and will be formally accepted for publication once it meets all outstanding technical requirements.

Kind regards,

Elsayed Abdelkreem, MD, PhD

Academic Editor

PLOS One

Reviewers' comments:

Reviewer's Responses to Questions

**Comments to the Author**

Reviewer #1: All comments have been addressed

Reviewer #2: All comments have been addressed

2. Is the manuscript technically sound, and do the data support the conclusions?

Reviewer #1: Yes

Reviewer #2: Yes

3. Has the statistical analysis been performed appropriately and rigorously?

Reviewer #1: Yes

Reviewer #2: Yes

4. Have the authors made all data underlying the findings in their manuscript fully available?

Reviewer #1: Yes

Reviewer #2: No

5. Is the manuscript presented in an intelligible fashion and written in standard English?

Reviewer #1: Yes

Reviewer #2: Yes

Reviewer #1: (No Response)

Reviewer #2: (No Response)

**Do you want your identity to be public for this peer review?** For information about this choice, including consent withdrawal, please see our Privacy Policy

Reviewer #1: **Yes:** Savas Tsikis

Reviewer #2: No

---

## [Editor Report · Acceptance letter]

PONE-D-25-35179R2

PLOS One

Dear Dr. Sun,

I'm pleased to inform you that your manuscript has been deemed suitable for publication in PLOS One. Congratulations! Your manuscript is now being handed over to our production team.

Kind regards,

on behalf of

Dr. Elsayed Abdelkreem

Academic Editor

PLOS One